# Adapting an Online Guided Self-Help CBT Programme Targeting Disordered Eating for Students in Aotearoa New Zealand: A Qualitative Study

**DOI:** 10.3390/nu16172905

**Published:** 2024-08-30

**Authors:** Alisa Mitlash, Charlene Rapsey, Gareth J. Treharne, Damian Scarf

**Affiliations:** 1Department of Psychology, University of Otago, P.O. Box 56, Dunedin 9054, New Zealand; gareth.treharne@otago.ac.nz (G.J.T.); damian@psy.otago.ac.nz (D.S.); 2Department of Psychological Medicine, Otago Medical School, P.O. Box 56, Dunedin 9054, New Zealand; charlene.rapsey@otago.ac.nz

**Keywords:** disordered eating, E-therapy, qualitative, university students, CBT

## Abstract

Disordered eating is a significant issue in university student populations. Currently, access to interventions is limited. Online interventions present an innovative way to increase accessibility to treatment for those in need. The current study explored how an online intervention for disordered eating (everyBody) could be modified to suit the needs of university students in Aotearoa New Zealand. Aotearoa New Zealand is a unique cultural context, with an indigenous population that has a high incidence rate of disordered eating, highlighting the need to adapt everyBody to the local context. Individual interviews were conducted with nine students currently at university in Aotearoa New Zealand, aged between 18 and 33 years old (five females, four males). Three first-order themes were identified using template analysis. The themes indicate that participants perceived the programme as acceptable and feasible for use with Aotearoa New Zealand’s university student population. Furthermore, the themes provide insight into potential adaptions to the programme to facilitate engagement and uptake. The suggested changes were largely consistent with previous research on E-therapy design (e.g., content length, therapeutic alliance), and also highlight changes specific to fit Aotearoa New Zealand’s cultural context. The findings have implications for universities and other funders deciding on services for students with disordered eating and eating disorders.

## 1. Introduction

Disordered eating can result in severe and debilitating consequences, particularly if left untreated [1]. University students are a vulnerable group for the experience of disordered eating and the development of eating disorders for a variety of reasons [2,3,4,5,6,7,8,9]. Indeed, a recent meta-analysis indicated around one in five students experience disordered eating, with a higher prevalence among women than men [1]. Thus, early intervention for disordered eating in university populations is imperative.

Currently, many students with disordered eating symptoms do not have access to treatment. For example, Eisenberg and colleagues [10] reported that only 20% of university students who screened positive for disordered eating symptoms received treatment within that year. Similarly, in Aotearoa New Zealand, a national survey indicated that just 1 in 4 individuals with Bulimia Nervosa and 1 in 10 individuals with Anorexia Nervosa contact treatment services at the age of symptom onset [11]. It is important to consider that these data date to a time before the increased popularity of social media [12,13,14,15]. Now, social media is commonly used and perpetuates unrealistic messages about body ideals, and eating and exercise habits, which in turn has a significant influence on disordered eating symptoms [16]. Still, these data indicate that there are treatment barriers for individuals with an eating disorder, including practical barriers (e.g., cost, transportation, inconvenience), previous negative experiences with treatment, a lack of knowledge, and stigma and shame. To reduce these barriers, we need innovations that increase the accessibility of treatment for those in need [17]. 

E-therapies are interventions that can be accessed through web-based apps and smartphones and present a fruitful opportunity for increasing treatment accessibility. A recent meta-analysis indicated that online therapies have a medium effect size in reducing disordered eating symptoms among university student populations [7]. One example of an effective E-therapy is StudentBodies [18,19]. StudentBodies is a guided online self-help programme for university students targeting disordered eating. The content of the programme is based on Cognitive Behavioural Therapy (CBT) for eating disorders [18,20].

A central component of StudentBodies is the guide or coach. E-therapy typically comes in guided (i.e., support from a trained online coach) and unguided formats (i.e., no support from an online coach). For eating disorder symptom reduction, there is evidence that a guided approach is superior to an unguided approach [19,21,22,23]. In addition, individuals with an online coach were significantly more satisfied with their treatment than individuals without an online coach [24]. Qualitative research mirrors the importance of including an online coach, whereby individuals with eating disorder symptoms report wanting guidance throughout the programme [25]. Thus, there is a meaningful benefit in using a guided format for eating disorder E-therapy [26], and it is recommended some level of face-to-face or virtual professional support is included to help maximise engagement and therapeutic outcomes [27]. 

### 1.1. Challenges of E-Therapy Interventions

It is well known that E-therapies can suffer from high attrition rates and suboptimal adherence [28,29]. The use of a guided format goes some way toward addressing these issues. Another strategy to reduce attrition is incorporating E-therapy end-user perspectives in adapting, designing, and developing interventions [27]. The use of qualitative methods can be particularly useful in capturing the needs, preferences and concerns of E-therapy end users [24,25]. For example, Nitsch and colleagues [30] used a mixed methods approach to explore engagement and usability issues within the StudentBodies programme in the United States. Participants completed one programme module and provided feedback on their experience. The central themes regarding usability and engagement issues pertained to navigation, content, layout, support, and engagement conditions. These results were used to make modifications, with follow-up data suggesting the changes increased the average usability score. 

Similarly, Yim and colleagues [31] explored individuals’ real-time and post-intervention experiences with everyBody Plus (including a group forum). EveryBody is a more recent version of StudentBodies, however the content largely remains the same, being based on CBT for eating disorders [31,32,33,34,35]. Participants reported that the programme was hard to navigate and that the modules felt too long. In addition, participants highlighted that the programme did not address everyone’s needs (e.g., those with mental health comorbidities), impacting their perceptions of the relevance and usefulness of the programme [30]. In response to the findings, Yim and colleagues [31] suggested specific changes for iterations of the programme, such as having a ‘book-marking’ function within the modules and improvement to accessing resources on the mobile version. Furthermore, they recommended emphasising to users that the programme cannot be a ‘one size fits all’ approach. 

### 1.2. Current Study

From the research reviewed above, it is evident that participant feedback is crucial to consider before undertaking large-scale RCTs as it helps reduce issues that may hinder uptake and engagement. Moreover, Yim and colleagues [31] emphasise that the feedback process is especially important for participants from marginalised groups such as people with marginalised ethnicities. Although informative, previous adaptations of StudentBodies were conducted in the United States [30] and the United Kingdom [31], potentially limiting their applicability to students in Aotearoa New Zealand. Moreover, there is a dearth of research assessing online interventions for disordered eating in Aotearoa New Zealand, let alone for university students. Furthermore, there is a need to consider the views of Māori university students as the Indigenous peoples as well as other marginalised university student groups. Therefore, we used a qualitative exploratory research design to assess whether the everyBody programme suits the cultural context of university students in Aotearoa New Zealand. Specifically, we aimed to broadly explore what modifications (for example, technical-related, design-related and Aotearoa New Zealand university student culture-specific issues) are needed to facilitate engagement and adherence. 

## 2. Materials and Methods

### 2.1. Design

The current study is a qualitative exploratory study. Individual semi-structured interviews were conducted with participants from a diverse range of ethnicities, two of whom identified as Māori, who were attending a university in Aotearoa New Zealand. The over-arching aim of the research was to investigate the suitability and acceptability of everyBody for students in Aotearoa New Zealand. The current study followed the guidelines of Brooks and King [36,37] for interview and analysis, DiCicco-Bloom and Crabtree [38] for interviewing, and Skivington et al. [39] for intervention evaluation. Ethics approval for the study was gained from the local university ethics committee (reference H21/100). 

### 2.2. Researcher Description

AM was the interviewer and analysed the data. She acknowledged her position in the research as White, female, and a young adult of similar age to most university students. She also acknowledged her experience as a trainee clinical psychologist and the primarily deficit-focused approach in this field. Furthermore, AM is an early career researcher; this was her second time conducting and analysing qualitative research. AM acknowledged her positionality and engaged in reflexivity, including keeping a journal in which she noted reflections and took field notes throughout the research process. 

### 2.3. Participants 

Nine university students (5 female, 4 male) aged between 18 and 33 years old (Mean = 20.6 years) participated. They reported a diverse range of ethnicities including NZ European (*n* = 3), Māori (*n* = 2), Asian (*n* = 2), and Other European (*n* = 2). Most of the participants identified as straight (*n* = 7), one identified as bisexual, and one reported being ‘not sure’ about their sexuality. Most participants had just completed their first year of university (*n* = 8), and the remaining participant was enrolled in post-graduate study. 

### 2.4. Recruitment

Purposeful sampling was used to select participants. Participants were recruited from a probability sample of university students who completed the World Mental Health International College Student Survey (WMH-ICS) [40] in 2023 and reported (a) having ever engaged in binge-eating behaviour and/or inappropriate compensatory purging behaviours for at least 3 months at some point in their life, and (b) consented to being informed about future research opportunities. The final sample of 315 students (female *n* = 232, male = 59) were all sent an invitation via Qualtrics (V. November 2023, Qualtrics, Prova; Provo, UT, USA) to participate in the current study, and this included detailed information about what would be involved in the study. Students were asked to provide their contact details if they were interested in participating in the study. The inclusion criteria were based broadly on criteria for an eating disorder in that participants had to have reported engaging in binge eating and/or inappropriate compensatory purging behaviours for at least three months at some point in their lives. In addition, participants had to be enrolled as a current student. 

Students who expressed interest were emailed an information sheet, demographics form, and consent form, and invited to attend an individual interview. The information sheet highlighted the aim of the present study, what would be involved in participating, and details of data collection and storage. Once consent was provided, an interview time was scheduled. Participants had the option of taking part in the interview in person or online; all the participants chose to do the interview online over Zoom (6.0.11, Zoom Video Incorporations, San Jose, CA, USA), and one participant chose to have their camera turned off.

The information power model [41], in conjunction with previous literature on similar study designs, and pragmatic considerations informed the sample size for the current study. This aligned with the research design, as we aimed to find out valuable and substantial contributions to how everyBody can best be adapted for students in Aotearoa New Zealand, rather than a complete description of the phenomenon of disordered eating among students. This would allow us to fulfill our practical aim of adapting the intervention. 

In the planning stages of determining sample size, we drew upon previous qualitative research which tested the usability of E-therapy for disordered eating [25,30,42,43]. In each cycle of their testing, they had a sample ranging from *n* = 4 to *n* = 12, and this was justified based on the justification that it takes 4–5 participants to detect 80% of usability issues and additional participants are increasingly less likely to reveal new information [44,45]. Therefore, when 9 interviews had been conducted (*n* = 5 female, *n* = 4 male) it was deemed that the data had sufficient information power. 

### 2.5. Online Intervention Programme Content

EveryBody is a digital, guided self-help, CBT intervention for disordered eating. EveryBody is based on the content of StudentBodies [31,32,33,34,35]. Access to the everyBody intervention content was provided to us by the research team who created StudentBodies [18]. The intervention includes six modules (see Table 1.), which are based on the core components and principles of CBT for eating disorders. The sequence of the modules follows the theoretical framework of CBT for eating disorders, whereby eating patterns and habits are addressed first, following skills to deal with maladaptive thoughts, feelings and behaviours [18,20]. At the end of each module, there is a summary to recap the covered content. The programme is offered through an online platform and each individual is assigned an online coach who provides timely individualised feedback and support.

### 2.6. Individual Interview Procedure 

Individual interviews were chosen over focus groups due to the sensitive nature of the topic of disordered eating and their potential to provide a more supportive environment for self-disclosure [46]. Participants were first briefed about the programme, followed by the use of a semi-structured interview schedule to elicit their thoughts. Interview topics and the creation of the interview schedule were informed by a review of the literature on online interventions for disordered eating, as well as broader literature about online mental health intervention design (see Appendix A). The interviews were held in a private research office over Zoom with just the interviewer and interviewee present. Data collection ran from mid-November 2023 to the end of December 2023. 

The individual interviews consisted of a discussion about module content and 23 pre-planned open-ended questions (see Appendix A). The interviews began with the primary researcher explaining the content of the online modules. A summary document was screen-shared with the participants, which included bullet point descriptions about module content (see Appendix A). At the end of each module, the interviewer asked the participants whether they had any thoughts, questions, or comments about the module. The participants were also invited to ‘think aloud’ during this part of the interview whereby they were encouraged to share any thoughts that came to their mind at any moment. The interviewer then asked the participants the pre-planned semi-structured questions within the broad domains of intervention-specific questions, challenges that students with disordered eating may be facing, and barriers and facilitators when seeking help for disordered eating. 

Participants were told that if they did not feel comfortable answering a question they did not have to, and that there are no right or wrong answers, it is just about what they think. Although every effort was made for the conversation to progress with a natural conversational flow, prompts were provided when the facilitator deemed it necessary to request additional information (e.g., if the participant was unsure how to answer a question, or for the participant to elaborate on their answer). Each participant was given a pseudonym to protect their identity; during the interview, they were given the option of choosing their own, otherwise, one was chosen for them that matched their gender and ethnicity. Participants were given a $50 (NZD) supermarket voucher for any costs incurred. The interviews lasted between 45 and 69 min (mean = 55 min). Most participants provided in-depth answers about their thoughts and some without much prompting. The interviews were audio-recorded, transcribed verbatim with the transcribing software Otter.ai (Version 1.24, Sam Liang and Yun Fu, Mountain View, CA, USA), and checked over for accuracy before analysis. 

### 2.7. Analytic Approach

Template analysis was used to conduct the qualitative analysis. Template analysis emphasises hierarchical coding, and thus a coding template is developed which summarises themes identified by the researcher as important and organises them in a meaningful manner [36,37]. The themes in template analysis represent recurrent features that are relevant to the research questions [44]. Template analysis was chosen as being in line with the research design as it is recommended for research questions of a practical nature and provides an approach that is clear, structured and systematic, yet offers flexibility to adapt to the needs of the study [36,37]. The goal was to identify themes in a meaningful and useful manner that could then be used for practical application in the everyBody intervention. 

Furthermore, the use of template analysis aligns with the critical realist epistemology applied during the collection, analysis, and interpretation of the data [36,37]. This approach was decided as appropriate because it allows for the university students’ narratives to be taken to reflect their expressed experience of the topic as their lived reality. However, it also considers that cultural context and social structure have a significant impact on the understanding and experience of mental health and disordered eating and that this may have an influence on the university students’ narratives [47].

Template analysis of individual interview transcripts was used in accordance with the 6 steps outlined by King [36]. As the research question was focused on the suitability and acceptability of everyBody, analysis was focused on a subset of the data which was relevant to answering the research question. The analysis began with a familiarisation phase where all interviews were transcribed and checked for accuracy, and field notes were taken. A priori themes (content, engagement and retaining users, marketing, coaching) were used as a loose guide in the preliminary coding phase. The a priori themes were based on the literature regarding online intervention design and the practical questions the research was asking. Coding was both deductive in the sense that a priori themes were used, and inductive in the sense that new codes were devised, modified, and deleted based on the participant’s narratives. This allowed for a structured approach while also not pre-empting the themes too much and potentially missing out on unique information from the participants

Coding and the iterations of templates were performed on Microsoft Word (version 16.88). All 9 interview transcripts underwent preliminary coding as the data set was relatively small. The codes were then organised into meaningful clusters that represented hierarchical relationships, which allowed for an initial template to be defined. This initial template was then applied to a subset of interviews (*n* = 3), and relevant modifications to the codes and themes were made, this was then repeated with several more subsets of data. 

The iterative process of trying out successive versions of the template and modifying and reapplying the template can continue for as long as seems necessary to allow a rich and comprehensive representation of the researcher’s interpretation of the data. This iterative process of modification and revision continued until no data relevant to the research question was left uncoded, and it was deemed to fit meaningfully into the template. The final version of the template was achieved on the seventh round of coding and contained three first-order themes (see Appendix A for the full template). An audit trail was kept of all developing templates and the final template was used for interpretation of the findings from the data. Data analysis was overseen by others in the research team (DS, CR, GT) to ensure the rigour of the process and final themes. 

## 3. Results

Three first-order themes were developed from the template analysis and are described in further detail below (see Appendix A for a complete code template, which demonstrates the coding hierarchy). The themes identified were (1) Acceptable programme content with consideration of contextual nuances, (2) Engagement and retaining users, (3) Ethics and management of personal information. Table 2 provides an overview of themes. Pseudonyms are used to protect the participants’ identity. 

### 3.1. Theme 1: Acceptable Programme Content with Consideration of Contextual Nuances

Overall, the content was perceived as relevant and suitable for students in Aotearoa New Zealand, and the participants’ discussion indicated a generally positive attitude toward the programme. 


*“I’m pretty happy with it [the programme] overall. I think it’s covered like all the bases really well. And it’s done in quite an engaging way. And yeah, I think it’s a really cool thing.”*
(Harry)

However, several participants were wary that the content may be perceived as *‘disingenuous’*, particularly if the language was something *‘we obviously don’t use here [in the cultural context of Aotearoa New Zealand]’*. This was particularly relevant to the delivery of the personal stories.


*“[referring to his thoughts on the personal stories] Yeah, I mean, I’ll be I’ll be completely honest with you. They sounded a little bit like ChatGPT [AI software] wrote them.”*
(Tom)

While there was consensus that personal stories may be beneficial in helping individuals not feel alone in their struggles, several participants cautioned that personal stories run the risk of creating negative comparisons.


*“…if you don’t have a personal story that you can sort of relate to or see similarities, then it could be even more isolating, like, ‘oh, do I really have an issue, if you know, my issue doesn’t compare to these other people’s’ kind of thing, because it could set up a comparison, especially with, with this sort of thinking, because most people with disordered eating, it does stem from comparison….”*
(Anna)

Furthermore, while the participants perceived an overall benefit of the programme, they recognised the diverse needs of individuals which the programme must consider, such as the multitude of ways that disordered eating presents, as well as the need to include education about problematic body ideals that span beyond the traditional thin-ideal. Drawing on the idea of diverse needs, the narratives of several participants indicated that some aspects of the programme might need to be modified or customised to suit the needs of the individual. 

For example, several participants stated that they would not be willing to engage in the social media challenges [Part of the Media Wellness module, the user is prompted to (a) post a photo on social media of something they love and (b) post an inspiring quote on social media].


*“[discussing the social media challenge]…there is personally, no way I would do that. Like, maybe if it was a different challenge, because I think having a challenge is really good.”*
(Bella)

Further, while some participants perceived the content as “[a] really good balance of having to kind of confront your own thoughts, but not forcing you to do anything that would make you too uncomfortable”, others suggested that the food planning/tracking maybe be perceived as ‘triggering’.


*“[discussing the self-monitoring logs]. Like once again, I think it could help a lot of people but there are also quite a few people, I think at least in my case, while I was going through like, the worst of it, that, that would be something that would be more destructive than positive for me. So I think it would just be important to like have a grain of salt on like, logging stuff about yourself, like for each person if that’s actually benefiting them, because I think that could be, yeah, could go either way.”*
(Jack)

The need for programme individualisation was also suggested with regard to communication with coaches.


*“I feel like that could be more up to the individual and how comfortable, they are on like, either calls or, like face to face, or like over text, I think it would be helpful to have it either on call or like face to face. But yeah, I think for some people, that might be too much. So I think it should just be more individualised.”*
(Jack)

Together, this indicates that practical measures such as screening for personal preferences and the provision of extra support/education resources could be useful to meet individual needs within the programme.

In addition, several participants raised issues around the cost of living and peer influences that were not covered in the programme content. They expressed that these issues have a significant influence on disordered eating in university students, indicating a potential need to acknowledge these issues within the content of the programme:


*“But the problem is, it’s just inflation with affordability. We can’t really solve it of course, but it would be great to mention.”*
(Jimmy)


*“…I felt like one of the main things that was bad for recovery was other people who had eating disorders, but like, weren’t fully aware of it, but talked about it as if it was normal…And it does affect like, basically everyone I’ve talked to with an eating disorder, and maybe even how to like approach people about that.”*
(Jack)

### 3.2. Theme 2: Engagement and Retaining Users

The participants’ narrative alluded to the reach of social media and its potential to promote the programme, particularly if the university were to get on board. Several participants perceived the student-led students association as a particularly valuable support resource for students and considered it a way to promote the programme.


*“[discussing how to make students aware of the programme] And services through like, [student association], like the support, because I thought people didn’t like use, [student] support at all. But then I came across my friends who like actually go through the support things. And I was like, Okay, this is like a big deal. So think, [student support] is a good area for this.”*
(Amrita)

Regarding continued use of the program, there was consensus among numerous participants that the ability to monitor and track one’s progress would be a key feature.


*“Giving like progress points, could be nice to make you want to keep going because if, you know, if it shows that you are actually improving, it might make you want to keep going.”*
(Anna)

Including such a feature may further allow for problem-solving and discussion with the coach to determine what may be going on for that individual. This would also facilitate timely (yet within realistic limits) feedback from the coaches. Several participants noted that it will be important for students to feel like their issues are validated and that they are provided adequate support for their ‘*in the moment*’ struggles. 

Furthermore, some participants discussed that notifications could be a useful way to increase programme adherence. It appeared that a busy student lifestyle was underlying this need for a reminder. More broadly, the student lifestyle was recognised as a potential barrier to engagement by several participants. One participant highlighted the need for the programme to acknowledge the constraints of student life. A particular constraint was time, and several participants cautioned that the module content may be too lengthy.


*“In regards to the programme, I think I would just say that acknowledging that being a student is hard. Like, our schedules are kind of all over the place…Because more often than not, you see all these programmes come up with all these like, like dietary plans, kind of thing that just aren’t realistic to fit into the lifestyle of a student. You know, telling you to make all these really fancy meals, when maybe sometimes you come home from school, and you’ve got maybe 20 min to make your dinner and you just can’t really be bothered, because you’re tired from the day. So I think just acknowledging that and just being aware of that….”*
(Anna)

Related to initial engagement, Anna described how the student lifestyle can result in the normalisation of disordered eating:


*[As students], you always hear about, you know, I didn’t I couldn’t afford to eat, so I just didn’t have dinner, or I always forget breakfast kind of thing. Like you don’t, you don’t really think it’s an issue as a student, because you hear so many people do it. It’s not quite disordered eating. Because it’s, you know, as students, it’s true, our schedules are so erratic, that sometimes you do skip a meal or something like that. But I think, you know, it does sort of make you think about oh, is, do I actually have disordered eating? Or is it just the student lifestyle kind of thing.”*


Further, Anna recognised that individuals who do not recognise that their behaviour is harmful are unlikely to seek help, saying, “*the hardest part in like eating disorders, is like accepting what you have is a problem.*” Thus, the ego-syntonic nature of disordered eating becomes a major barrier to engagement. However, several participants thought the availability of the programme could help facilitate the “*acceptance*” process. 


*“…I think it [the programme] will help people understand and realise like, what they’re going through, and then it’s not wrong to like, seek help and like, talk about it.”*
(Amrita)

At the same time, engaging in the programme could be perceived as “*overwhelming*” for someone in the acceptance stage. Several participants suggested making a “bite-sized” version of the programme could mitigate feeling overwhelmed. A briefer version of the programme could facilitate buy-in and may also be useful in overcoming the issues that participants described regarding module length. 


*“Having a shorter version just to dip your toes in and see, you know, ‘is this something I’m interested in’, ‘is it something that can help me’…could be useful.”*
(Anna)

### 3.3. Theme 3: Ethics and Management of Personal Information

This theme illustrates the importance of confidentiality and privacy. Participants explained that the advantage of an online program is anonymity, which may act as a gateway for disordered eating help-seeking.


*“…help things or programs, kind of like this [online ED programme], where you don’t necessarily have to go see a doctor…I think those are good because then you’re not worried that like, the whole universe is gonna find out.”*
(Bella)

Another participant emphasised that anonymity may be particularly useful for men who are experiencing disordered eating.


*“I guess there still is a bit of a stigma. So I think kind of like an online program like this might be good because you can kind of do it without having to sit down in front of anybody or open up and be vulnerable. And yeah, I just think for a lot of people, it’s a really uncomfortable kind of topic to talk about, especially for like males. It’s not something that like most people like talking about or opening up about. So I think, yeah, having an option that can be done online, or just having a coach that talks to you virtually, is really, really helpful.”*
(Harry)

Furthermore, there was consensus among the participants that the coaches must have adequate training for the intervention to be safe and effective, thus highlighting the importance of a thorough protocol. However, there were differences among the participants as to whether they would prefer a coach who was more professional or someone who was more relatable.


*“I feel like a good mix, or that they do have that kind of background, but they’re also more relatable as well. It’s like, just like a happy medium. Because sometimes if you talk to someone who’s like, fully trained and a doctor, it’s all medical. And you’re kind of cool yep, whatever. But when it’s someone you can relate to a little bit more, you tend to take in a bit better. So a wee happy medium between the two.”*
(Lucy)

For some participants, relatability with their coach was paramount:


*“[discussing her thoughts on coaches] Yeah, that kind of whole match of, you know, figuring out whether you feel like, you can relate to this person and feel like they would understand you in the way that you kind of want to be understood. It’s really a big thing.”*
(Rose)

Together, the diverse range of responses to what students expect in a coach links back to the idea of diverse needs. This leads to the question of what can be practically done to cater to student’s needs, while also remaining within safe professional boundaries. For example, the current system has a function to ‘choose your coach’, so to help participants choose the right coach for them, this could be further facilitated by including a ‘get to know your coach’ video, screening for coach characteristic preferences, describing the coaches cultural background, and using prompts in an introduction session to create a point of connection between coach and participant.

## 4. Discussion

The current study highlights that, overall, the content of everyBody was perceived as acceptable by university students in Aotearoa New Zealand. However, the diverse students who participated also alluded to changes to the content that would help increase engagement and uptake. There were suggested changes that were specific to the cultural context of university students in Aotearoa New Zealand, such as the recognition of cultural ideal body types and the use of terminology that students typically use in Aotearoa New Zealand. In addition to findings specific to the cultural context of Aotearoa New Zealand, overall, our findings are largely consistent with several other studies exploring e-therapy design; we found factors such as language, content length, ethics/management of personal information, progress monitoring, therapeutic alliance, and notifications are important to consider in e-therapy design [25,30,31,42,48,49,50,51,52]. Incorporating such feedback is likely to result in higher engagement through greater usability, perceived usefulness, and user satisfaction, which together can result in increased clinical impact [53]. The potential implications and applications of the findings in relation to wider trends in e-therapy design are discussed below. 

The current study further confirms the difficulties with a ‘one size fits all’ approach to E-therapies for eating disorders [31]. The participants highlighted the importance of being able to customise the program to suit individual needs and preferences. Furthermore, they discussed the need for the content to match the current cultural and social context of university students in Aotearoa New Zealand, this included having examples that are relevant to Aotearoa New Zealand (e.g., using common language terms ‘hall’ instead of ‘dorm’), and referring to body ideals that span beyond the thin ideal. 

In addition, Māori as the Indigenous peoples, and Pacific peoples, are likely to have a different perception of ideal body image [54,55]. For example, Māori perceive their cultural ideal body as larger than that of the Western thin ideal for women [54,56]. Similarly, Pacific people describe an ideal body image that is larger than that of the traditional Western ideal [55,57]. It is suggested that Pacific peoples’ body ideals are closely linked to gender roles within Pacific culture. Thus, Pacific ideal body perceptions tend to focus on functionality (e.g., childbearing, providing for family/community), as opposed to aesthetic purposes [55]. 

While such populations are still negatively influenced by Westernised beauty standards [56,57], Māori and Pacific cultural ideals and the values behind them could be used to counteract the consequences [58]. Indeed, a study with young Māori and Pacific wāhine (women) reflected that this group of young people drew on their cultural knowledge and cultural traditions to create strengths-based views of their bodies, which subsequently improved their relationships with their bodies [57]. Therefore, such cultural strengths-based views should be drawn upon in the context of disordered eating e-therapy to facilitate the development of positive body image. Inclusion of such content would require a personalised approach and therefore further draws importance to the integral role of the coach in doing so. 

Furthermore, the results highlight that there can be tension between what end users want and what is prescribed within the intervention, which could create a barrier to treatment adherence. In this study, some participants alluded to the idea that self-monitoring, in particular food records and meal planning, may not be appropriate or useful to include within the intervention. Some participants described self-monitoring and regular eating in the current study as *‘triggering’ referring to notions of antecedents for uncomfortable emotions and thoughts, which is common among people going through therapy with an eating disorder* [59]. This is also reflected in previous qualitative research which explored individuals’ perspectives of eating disorder treatment, whereby *essential core* components of treatment (such as self-monitoring and structured assistance with eating) received a significant amount of negative comments [60]. 

However, self-monitoring is described as a cornerstone of CBT for eating disorders [59,61,62,63], and several studies indicate that self-monitoring is effective at reducing eating disorder symptomology [64,65,66,67]. Swain-Campbell et al. [60] concluded that while it is important to recognise the distress that may be experienced from treatment, it alone does ‘not substantiate the grounds for seriously reviewing the inclusion of core components’. However, there are ways in which these feelings of distress can be managed without de-emphasising the use of core treatment components [59], thus lending to several practical recommendations for the implementation of everyBody (see Appendix A).

Adjacent to these practical recommendations, it will be important to bear in mind that individual differences may contribute to engagement in intervention and its outcomes. For example, the individual’s readiness to change [68], symptom severity [69], and presentation of symptomology (e.g., an ego-syntonic value that may be placed on their current behaviour) [59,67] have all been found to impact intervention engagement. Such individual differences may contribute to explaining why some participants in the current research indicated that they thought the core components of the program provided the right amount of challenge for them and saw the content as beneficial, while other participants thought that it should not be included. Future research could focus on determining the effectiveness of specific treatment components such as self-monitoring within different presentations of disordered eating.

Additionally, Graham and colleagues [53] discussed the need for e-therapy for disordered eating to be designed with equity in mind. In Aotearoa New Zealand, equitable access to culturally responsive treatment should be a priority for Māori, Pacific peoples, and those who identify with queer sexualities. Often traditional mental health care does not align with the needs of populations such as Māori, Pacific peoples, and queer sexualities and thus resulting in higher rates of disordered eating and lower rates of treatment access [70,71,72,73]. While the current study has captured the perspectives of two Māori individuals, research addressing ongoing inequities in treatment access within Aotearoa New Zealand remains important [74,75,76,77].

## 5. Conclusions

This qualitative study has provided preliminary evidence of the acceptability and feasibility of everyBody for university students in Aotearoa New Zealand, and demonstrates key considerations for users across the themes of acceptable programme content with consideration of contextual nuances, engagement and retaining users, and, ethics and management of personal information. With participants’ feedback having informed programme improvements [53], the next step is to conduct a feasibility pilot of everyBody within a sample of university students from Aotearoa New Zealand who are experiencing disordered eating. The ultimate goal of this research is to bridge the treatment gap [12,23] for students experiencing disordered eating in Aotearoa New Zealand. 

## Figures and Tables

**Table 1 nutrients-16-02905-t001:** Content of original everyBody online intervention.

Module	Content
Module 1 Balanced eating	Introduction, setting personal goals, psychoeducation on eating disorders, regular eating and balanced eating habits, meal planning and meal tracking, mindful eating activity
Module 2 Coping well	Psychoeducation about emotional triggers, dealing with triggers activity, mindfulness activity
Module 3 Thinking well	Psychoeducation about automatic thoughts, the thin ideal and social comparison, cognitive restructuring, monitoring logs, behavioural experiments, mindfulness activity
Module 4 Body wellness	Psychoeducation about body image and healthy exercise, values, decreasing shape checking and body avoidance, assessing exercise habits
Module 5 Media wellness	Psychoeducation about the media, social media challenges
Module 6 Relationship wellness and relapse prevention	Psychoeducation about interpersonal relationships, interpersonal communication activity, relapse prevention plan

**Table 2 nutrients-16-02905-t002:** Presentation of themes and overview of their content.

Themes in Hierarchical Order	Overview of Theme Content
**1.** **Acceptable programme content with consideration of contextual nuances** ***1.1.*** ** *Content is relevant, relatable and acceptable* ** 1.1.1. But make sure it sounds genuine *1.1.1.1.* *Important to be mindful about how personal stories are used* 1.1.1.1.1. Risk of creating comparison 1.1.1.1.2. Wording needs to suit the Aotearoa New Zealand context ***1.2.*** ** *Diverse needs and customisation* ** 1.2.1. The social media challenge is not for everyone *1.2.1.1.* *Not everyone uses social media this way* *1.2.1.2.* *The dangers of social media* *1.2.1.3.* *Alternative challenge* 1.2.2. Challenging but not ‘triggering’ 1.2.3. Logbooks and food diaries might not be suitable for some *1.2.3.1.* *Support resources* 1.2.4. Providing a choice for coach communication *1.2.4.1.* *Level of formality* *1.2.4.2.* *Stage of their journey* 1.2.5. It’s about more than the thin ideal *1.2.5.1.* *Body positivity* *1.2.5.2.* *Masculine ideal* ***1.3.*** ** *Addressing important issues* ** 1.3.1. The cost of living and food anxiety 1.3.2. Peer influences *1.3.2.1.* *‘How can I support my peers’*	This is about what participants thought about the content of the modules. This was a big theme as a lot of the discussion was focused on the module content. Overall, the participants thought the content was relevant and suitable. However, participants also discussed content that may need modification, and new content that may need to be included.
**2.** **Engagement and retaining users** ***2.1.*** ** *Facilitators* ** 2.1.1. Ability to see and monitor progress 2.1.2. Coach responsivity *2.1.2.1.* *Sensitivity* *2.1.2.2.* *Follow-up* 2.1.3. Notifications ***2.2.*** ** *Barriers* ** 2.2.1. The student lifestyle *2.2.1.1.* *Module length* 2.2.1.1.1. Feeling bored 2.2.2. It’s hard to accept it as a problem *2.2.2.1.* *Potential to promote awareness* *2.2.2.2.* *Trialling the programme* ***2.3.*** ** *Marketing of online intervention* ** 2.3.1. Easy accessibility 2.3.2. Social media 2.3.3. Around the university *2.3.3.1.* *Student Association*	This theme is about what participants perceived may act as facilitators or barriers in student’s engagement with the programme. They also suggested how the programme could be promoted to students.
**3.** **Ethics and management of personal information** ***3.1.*** ** *Confidentiality and privacy* ** ***3.2.*** ** *Safety and training* ** 3.2.1. The spectrum from relatable to professional *3.2.1.1.* *The need to be able to relate*	Confidentiality and privacy were important to the participants. They perceived the anonymity of the programme as an advantage. However, participants demonstrated differences in preference for the level of professionalism and relatedness within their coach.

## Data Availability

The data presented in this study are not readily available due to ethical restrictions. Requests to access the data set should be directed to the corresponding author.

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
