# Peer review of "Adapting an Online Guided Self-Help CBT Programme Targeting Disordered Eating for Students in Aotearoa New Zealand: A Qualitative Study"

_nutrients, 2024, doi:10.3390/nu16172905_

Round 1

Reviewer 1 Report

Comments and Suggestions for Authors

paper entitled "Adapting an online guided self-help CBT programme targeting disordered eating for students in Aotearoa New Zealand: A qualitative study"

is interesting, here are some comments, questions, and recommendations:

as the abstract noted that there are various culture specific nuances within online intervention - could the author/s elaborate on this

- culture specific in terms of ethnicity? or countries? wherein the online intervention is used?

or West VS East

- or culture specific to different generations of students? GenZs?

- or international students samples?

the author/s also mentioned about studentbodies (also an online intervention?)

in what ways are the current software used different or similar with the one in the literature review?

should clarify the issues that are culture specific, because it would seem that most issues are technical related (navigation/user interface issues) or design related (too long)

the issue would now be if the current software design addresses a multiculture audience

- a side note - within the Table S1 - "Use language more suitable to cultural context." the author/s noted - E.g. ‘first year’ instead of ‘freshman year’. 

this might not be a cultural issue

is there an overall theoretical background framework for the design of the online intervention?

specifically the sequence of the modules

findings seems fine, however, might not reflect the title, might consider "Experiences of an......"

for the discussions and conclusions - should also go back to the issue of culture, which was noted as an issue with the reception of online intervention

Author Response

Dear reviewer,

Thank you very much for taking the time to review this manuscript. Please find the detailed responses below and the corresponding revisions/corrections highlighted in yellow in the re-submitted files.

Comment 1: as the abstract noted that there are various culture specific nuances within online intervention - could the author/s elaborate on this

- culture specific in terms of ethnicity? or countries? wherein the online intervention is used?

or West VS East

- or culture specific to different generations of students? GenZs?

- or international students samples

Response 1: Thank you for pointing this out. We agree that what we mean by culture specific needed to be clarified. Therefore, we have made it clear in both the abstract and the main text that we are focusing on the specific cultural context of university students within Aotearoa New Zealand. You can find this clarification added on line 16,17; page 1; abstract, line 103-105 and line 107-110; paragraph 7; page 3.

Comment 2a:

the author/s also mentioned about studentbodies (also an online intervention?)

 Response 2a:

We have now explicitly stated that StudentBodies is an online intervention. It now reads:

‘One example of an effective E -therapy is StudentBodies [18, 19]. StudentBodies is a guided online self-help programme for university students targeting disordered eating’. Please see line 55-57; paragraph 3; page 2.

Comment 2b in what ways are the current software used different or similar with the one in the literature review?

Response 2b:

EveryBody (current software) is a new version of StudentBodies that has been designed by the same research team (Fitzsimmmons-Craft et al. 2020). EveryBody is based off the same core components of StudentBodies. We have clarified this in the introduction which now reads:

‘EveryBody became the newer version of StudentBodies, however the content remains the same, being based on CBT for eating disorders’, line 84-86; paragraph 6; page 2.

It has also been clarified in the methods and now reads:

‘EveryBody is based on the content of StudentBodies [31-35]. Access to everyBody intervention content was provided to us by the research team who created StudentBodies’. Please see line 174,175; paragraph 16; page 4.

Comment 3: should clarify the issues that are culture specific, because it would seem that most issues are technical related (navigation/user interface issues) or design related (too long)

the issue would now be if the current software design addresses a multiculture audience

Response 3: We have now clarified that the current study is focusing on all types of feedback such as technical or related issues, as well as issues pertaining to cultural context. We have added in the word ‘broadly’ into our aims to emphasize this point.

Line 107-110; paragraph 7; page 3; now reads: ‘Specifically, we aimed to broadly explore what modifications (for example, technical related, design related and Aotearoa university student culture specific related issues)’.  

We expected to see some similarities with previous research, and therefore the word ‘nuances’ was chosen in the abstract to reflect subtleties in the experience of the intervention in the cultural context of university students in Aotearoa New Zealand.

Comment 4: - a side note - within the Table S1 - "Use language more suitable to cultural context." the author/s noted - E.g. ‘first year’ instead of ‘freshman year’. 

this might not be a cultural issue

Response 4: In Aotearoa New Zealand we do not use the term ‘freshman’ and therefore we have retained the wording ‘first year’ as a good example of a cultural issue of relevance to wording in interventions.

Comment 5: is there an overall theoretical background framework for the design of the online intervention? Specifically the sequence of the modules

Response 5: We have now made it clearer that the online intervention is based on Cognitive Behavioural Therapy (CBT) for eating disorders.  The sequence of the modules is therefore based on the components of CBT for eating disorders. We have, modified the methods section to emphasize this point. It now reads:

‘The intervention includes six modules (see table 1.), which are based on the core components and principles of CBT for eating disorders. The sequence of the modules follows the theoretical framework of CBT for eating disorders whereby eating patterns and habits are addressed first, following skills to deal with maladaptive thoughts, feelings and behaviours.’ Line 176-180, paragraph 16; page 4.

Comment 6: findings seems fine, however, might not reflect the title, might consider "Experiences of an......"

Response 6: We have kept the title as it because the study focuses on what can be adapted in the intervention which is to be informed by feedback of user experiences.

Comment 7: for the discussions and conclusions - should also go back to the issue of culture, which was noted as an issue with the reception of online intervention

Response 7: We have again clarified the cultural context of university students in Aotearoa New Zealand, please see line 480-482, first paragraph in discussion, page 11. In the discussion we have written about how cultural difference can impact the reception of an online intervention and what could be done in the context we are focusing on, please see line 494-515; third paragraph in discussion; page 11. We have also expanded on the issue of culture and emphasized specific cultural groups that should be focused on in future research, line 546-554; last paragraph in discussion; page 13.

Reviewer 2 Report

Comments and Suggestions for Authors

This paper presented the investigation of the adaption of an online programme to monitor the disordered eating for university students. The findings may help the universities prevent eating disorder for students. However, the manuscript needs some necessary revisions.

1. More findings should be presented in abstract section.

2. The introduction section was too tedious. The content should be simplified. The authors should focus on the necessity and importance of this study. The introduction of background and previous investigations should be shortened.

3. How to design this investigation? What guidelines or methods were referred? Please explain.

4. The presentation of the results or the findings should be reorganized. The authors used a lot of texts to described the results. I suggested that some tables or figures can present the findings more clearly and visually.

5. Please carefully check the format of references based on the journal’s guidelines.

Author Response

Dear reviewer,

Thank you very much for taking the time to review this manuscript. Please find the detailed responses below and the corresponding revisions/corrections highlighted in blue in the re-submitted files.

Comments 1: More findings should be presented in abstract section.

Response 1: Thank you for pointing this out. We have added more results to the abstract. It now reads:

Disordered eating is a significant issue in university student populations. Currently access to intervention is limited. Online interventions present an innovative way to increase accessibility to treatment to those in need. The current study explored how an online intervention for disordered eating (everyBody) could be modified to suit the needs of university students in Aotearoa New Zealand. Aotearoa New Zealand is a unique cultural context, with an indigenous population that has a high incidence rate of disordered eating, highlighting the need to adapt everyBody for the local context. Individual interviews were conducted with 9 students currently at university in Aotearoa New Zealand, aged between 18 and 33 years old (5 females, 4 males). Three first-order themes were identified using template analysis. The themes indicate that overall, participants perceived the programme as acceptable and feasible for use with Aotearoa New Zealand’s university student population. Furthermore, the themes provide insight into potential adaptions to the programme to facilitate engagement and uptake. The suggested changes were largely consistent with previous research on E-therapy design (e.g., content length, therapeutic alliance), and also highlight changes specific to Aotearoa New Zealand’s cultural context. The findings have implications for universities and other funders deciding on services for students with disordered eating and eating disorder.’

We could not add too much as we are mindful to stay in the abstracts word count. See line 21, 23-25, page 1, abstract.

Comments 2: The introduction section was too tedious. The content should be simplified. The authors should focus on the necessity and importance of this study. The introduction of background and previous investigations should be shortened.

Response 2: We have revised the introduction to be more streamlined whilst retaining the relevant background on past studies as these are needed to demonstrate the current state of the literature and justify our study as a way of expanding on existing knowledge. Paragraph 1, 3 and 4 (page 1 and 2) have been shortened.

Comments 3: How to design this investigation? What guidelines or methods were referred? Please explain.

Response 3: We have added clarification on the methodological guidelines we followed for interviewing, analysis and intervention evaluation. It now reads: ‘The current study followed the guidelines of Brooks and King [36, 37] for interview and analysis, DiCicco-Bloom and Crabtree [38] for interviewing, and Skivington et al. [39] for intervention evaluation.’ – see line 117-119, paragraph 8, page 3.

Comments 4:  The presentation of the results or the findings should be reorganized. The authors used a lot of texts to described the results. I suggested that some tables or figures can present the findings more clearly and visually.

Response 4: We have reorganized the results to make them clearer. We have added in a table at the beginning of the results providing an overview of the themes (see page 7). We have made the quotes italicized and created space between them to make them more clear (see pages 8, 9, 10, 11). We have kept the quotes within the results section as they are important in demonstrating evidence. King also recommends presenting template analysis results in such a way.

Comments 5: Please carefully check the format of references based on the journal’s guidelines.

Response 5: The references have been checked to ensure they follow the ACS guidelines.

Round 2

Reviewer 1 Report

Comments and Suggestions for Authors

After going over the point by point revisions made by the researcher, the paper is now clearer and is adequate for acceptance